# Transgenerational Plasticity Enhances the Tolerance of Duckweed (*Lemna minor*) to Stress from Exudates of *Microcystis aeruginosa*

**DOI:** 10.3390/ijms252313027

**Published:** 2024-12-04

**Authors:** Gengyun Li, Tiantian Zheng, Gang Wang, Qian Gu, Xuexiu Chang, Yu Qian, Xiao Xu, Yi Wang, Bo Li, Yupeng Geng

**Affiliations:** 1Ministry of Education Key Laboratory for Transboundary Ecosecurity of Southwest China, Yunnan Key Laboratory of Plant Reproductive Adaptation and Evolutionary Ecology, Institute of Biodiversity, School of Ecology and Environmental Science, Yunnan University, Kunming 650504, China; 2College of Landscape and Horticulture, Yunnan Agricultural University, Kunming 650201, China; 3Yunnan Collaborative Innovation Center for Plateau Lake Ecology and Environmental Health, College of Agronomy and Life Sciences, Kunming University, Kunming 650214, China; 4Great Lakes Institute for Environmental Research, University of Windsor, Windsor, ON N9B 3P4, Canada

**Keywords:** duckweed, cyanobacteria, MaE, transgenerational plasticity, transcriptional memory

## Abstract

Transgenerational plasticity (TGP) refers to the influence of ancestral environmental signals on offspring’s traits across generations. While evidence of TGP in plants is growing, its role in plant adaptation over successive generations remains unclear, particularly in floating plants facing fluctuating environments. Duckweed (*Lemna minor*), a common ecological remediation material, often coexists with the harmful bloom-forming cyanobacterium *Microcystis aeruginosa*, which releases a highly toxic exudate mixture (MaE) during its growth. In this study, we investigate the TGP of duckweed and its adaptive role under stress from MaE during the bloom-forming process. We found that exposure to MaE induces significant phenotypic plasticity in duckweed, manifested by alterations in morphological, physiological, and transcriptomic profiles. Specifically, MaE exposure significantly affected duckweed, promoting growth at low concentrations but inhibiting it at high concentrations, affecting traits like biomass, frond number, total frond area, and photosynthetic efficiency. Additionally, the activities of antioxidant enzymes, together with the levels of proline, soluble sugars, and proteins, are elevated with increasing MaE concentrations. These plastic changes are largely retained through asexual reproductive cycles, persisting for several generations even under MaE-free conditions. We identified 619 genes that maintain a ‘transcriptional memory’, some of which correlate with the TGP-linked alterations in morphological and physiological traits in response to MaE stress. Notably, progeny from MaE-exposed lineages demonstrate enhanced fitness when re-exposed to MaE. These results enhance our comprehension of the adaptive significance of TGP in plants and suggest feasible approaches for utilizing duckweed’s TGP in the bioremediation of detrimental algal blooms.

## 1. Introduction

Phenotypic plasticity allows organisms to adapt to changing environments by producing different phenotypes from the same genotype [1,2]. Transgenerational plasticity (TGP) is a specific form of phenotypic plasticity where parental environmental experiences influence offspring phenotypes [3,4,5]. This mechanism is particularly important for clonal plant species, as their offspring share the same genetic makeup and often inhabit similar environments as the parent plant [6,7]. TGP can potentially prepare clonal progeny for environmental challenges similar to those experienced by their parents, enhancing their survival and fitness [4,8,9]. In particular, clonal reproduction may facilitate more efficient transmission of parental environmental effects to offspring by bypassing genetic recombination and epigenetic resetting associated with sexual reproduction [10]. Although many studies have documented TGP in plant species [5,8], its ecological and evolutionary significance remains to be fully understood.

Comparing gene expression across generations offers a powerful tool to address how TGP affects the adaptation and success of clonal plant populations in changing environments. Parental environmental effects can be transmitted to offspring through various pathways [5], including epigenetic modifications or various metabolites, which are capable of moderating gene expression during progeny development and affect the incidence of TGP [8]. By examining changes in gene expression patterns between generations, researchers can identify specific genes and pathways that are altered in response to parental environmental conditions [5,6,11,12]. For instance, parents exposed to stressful conditions can frontload the protective transcripts in offspring by upregulating antioxidant and stress response genes, preparing them for similar stresses [13].

*Microcystis aeruginosa* is one of the most pervasive bloom-forming cyanobacteria worldwide, which can rapidly proliferate under favorable conditions, leading to massive blooms in freshwater ecosystems [14,15]. *M. aeruginosa* is known to produce microcystin, a potent cyanotoxin that is predominantly confined within cyanobacterial cells and is released into aquatic ecosystems when the cells die and lyse [16]. Numerous studies have focused on the deleterious effects of microcystin on aquatic organisms. However, it is crucial to recognize that during the proliferation of *M. aeruginosa*, a highly toxic exudate mixture, henceforth referred to as MaE, is also secreted. This complex of compounds can influence other aquatic organisms at an earlier stage, posing significant threats to biosecurity, food webs, and public health even before the bloom explosion [17,18,19]. Recent studies have indicated that MaE is a compound composed of more than 300 chemical substances, including a large number of lipids, organic heterocyclic compounds, organic acids, benzene compounds, organic oxygen compounds, phenylpropanoids, and organic nitrogen metabolites, among which the toxicological effects of various compounds have been confirmed [19,20]. Several studies have revealed the toxic effects of MaE on various aquatic organisms, including *Potamogeton malaianus* [16], *Potamogeton crispus* [21], *Sinocyclocheilus graham* (an endangered fish [22]), and the biofilm microbial community on the leaves of *Vallisneria natans* [23]. Compared with the above species that are highly sensitive to *MaE*, duckweed seems to have greater tolerance [18]. In field observations, duckweed (*Lemna minor,* Lemnaceae) is often found coexisting with toxic cyanobacteria blooms and has been proposed as a model plant for bloom control and water pollution remediation [24].

Duckweed is a small aquatic plant known for its remarkable clonal reproduction, allowing it to achieve rapid population growth. In many cases, the entire duckweed population may originate from a single maternal genotype [25], highlighting the efficiency of its asexual reproduction strategy. Despite the low levels of genetic variation within populations, duckweed may exhibit exceptional adaptability, thriving in diverse and variable environments with water bodies that have highly fluctuating physical and chemical properties [26]. Duckweed’s exceptional environmental adaptability makes it an ideal model for various ecological and evolutionary studies [27], and it is widely used for the phytoremediation of a wide range of pollutants [24]. Phenotypic plasticity plays a crucial role in enabling duckweed to rapidly adapt to environmental changes [28]. Given the significance of plasticity for plant adaptation, it can be speculated that phenotypic plasticity, including TGP, likely plays a role in duckweed’s process of adapting to various environmental stresses. However, how these mechanisms operate, particularly how TGP aids duckweed and its progeny in developing adaptive responses to dynamic aquatic environmental stresses such as MaE stress, has yet to be elucidated.

In this study, we investigate the role of phenotypic plasticity, particularly concerning TGP, in duckweed’s tolerance to MaE using *M. aeruginosa*. We first assessed phenotypic plasticity at various levels in duckweed, including morphological, physiological, and gene expression, in response to various MaE concentration treatments. Then, we examined the effect of parental exposure to MaE on the TGP of clonal offspring that were produced under MaE-free conditions. We further assess the adaptive significance of TPG using a reciprocal transplant design, which allows us to test whether TPG leads to better performance for clonal offspring when encountering the re-appeared stress via MaE.

## 2. Results

### 2.1. TGP of Duckweed to Different Concentrations of MaE in P1–P3

We examined phenotypic plasticity in duckweed, spanning from P1 treated with MaE to stress-free P2–P3 offspring and to P4 generations re-exposed to MaE (Figure 1). Morphological traits exhibited varying responses to different concentrations of MaE. In the P1 stage, several fitness traits, including the biomass, number of leaves, and total leaf area, showed a ’low promotion, high inhibition’ pattern, with significant increases under low MaE conditions (*p* < 0.05), followed by decreases under medium and high MaE conditions (Figure 2A–D). The length of the root significantly increased with the rise in the MaE concentration (*p* < 0.05) (Figure 2E). Photosynthetic traits responded differently; baseline fluorescence (F0) decreased significantly at low MaE in P1 (*p* < 0.05) but increased above control levels at high concentrations (Figure 2F). The maximum fluorescence (Fm), the ratio of variable to maximum fluorescence (Fv/Fm), and the contents of chlorophyll a and chlorophyll b also demonstrated a ‘low promotion, high inhibition’ response (*p* < 0.05) (Figure 2G–J). Multiple physiological indicators, such as antioxidant enzyme activities (SOD, POD, and APX), proline, soluble sugar, and protein, showed a significant increase (*p* < 0.05) as the MaE treatment concentration increased (Figure 2K–P).

The response of multiple traits to a high concentration of MaE can continue through to P2 and even P3, yet as cultivation time extends, the trait differences between the P1S progeny and the P1CK progeny gradually diminish. For the P1S progeny, all the traits we assessed still retained their differences (*p* < 0.05) from the P1CKprogeny during the P2 phase after the cultivation conditions were shifted back to CK (Figure 3). However, in comparison to the P1 phase, the extent of variation in several traits was attenuated, such as the total leaf area (Figure 3B), root length (Figure 3C), soluble sugar content (Figure 3J), and activities of antioxidant enzymes (Figure 3L–O). By the P3 phase, differences in some traits ceased to exist, like the number of leaves (Figure 3A), total leaf area (Figure 3B), content of chlorophyll a (Figure 3D), and Fv/Fm ratio (Figure 3H). Nevertheless, certain traits continued to show significant differences (*p* < 0.05), including root length (Figure 3C); content of proline, soluble sugar, and protein (Figure 3I–K); and SOD enzyme activity (Figure 3N).

### 2.2. Effects of P1 Environment on P4 Populations

We examined whether the parental environment in the P1 stage affected the performance of P4 offspring when re-exposed to high concentrations of MaE. This was achieved by comparing the trait expression of P4 offspring from different parental sources (P1CK and P1S) under control (P4CK) and high MaE (P4S) conditions (Figure 4). Exposure of the P1 stage to MaE had no significant impact on root length, CAT, and APX activity (Figure 4J) in P4 individuals when re-exposed to MaE (P1CK–P4S vs. P1S–P4S, with LSD test *p*-value > 0.05). However, P1 MaE exposure affected the expression of other traits in P4 when re-exposed to MaE. Based on the direction of the reaction norm, these traits can be classified into two types: (1) Weaken traits: pre-adapted to MaE stress, showing less of a decrease in P1S–P4S than P1CK–P4S (*p* < 0.05) (e.g., biomass, leaf number, leaf area, chlorophyll b content, Fm, and Fv/Fm) (Figure 4A–D,F,H,I). (2) Strengthen traits: increased expression under MaE stress, showing a greater increase in P1S–P4S than P1CK–P4S (*p* < 0.05) (e.g., POD and SOD activity, content of proline, soluble sugar, and soluble protein) (Figure 4K–O). The expression trends of F0 and chlorophyll a in P1–P4 were inconsistent, so they were not classified into either of these two categories.

### 2.3. RNA Sequencing of Duckweed

On average, each RNA-seq library yielded 6.65 G of clean bases. The average error rate for each library was 0.03, with mean Q20 and Q30 values of 97.00% and 92.12%, respectively. The GC content ranged from 52.99% to 53.79% (Appendix A).

Regarding sequences mapped to the reference genome, the unique mapping rate ranged from 89.86% to 91.19%, with an average of 90.58%. In the context of alignment to the reference genome sequences, an average of 81.32%, 5.12%, and 13.56% was, respectively, mapped to exon, intron, and intergenic regions (Appendix A). A total of 25,604 transcripts were obtained in the end, with 3222 transcripts generated through new transcript assembly.

### 2.4. Characterization of MaE-Responsive Genes and TMGs

Differential expression genes (DEGs) were identified by comparing the conditions S, P1S–P2CK, P1S–P3CK, and P1S–P4S to CK. In the P1 stage, the comparison between S and CK identified 3870 DEGs (comprising 1578 upregulated and 2112 downregulated genes). In the P2 stage, the comparison between P1S–P2CK and CK revealed 2595 DEGs (consisting of 1428 upregulated and 1167 downregulated genes). For the P3 stage, the comparison between P1S–P3CK and CK yielded 1580 DEGs (including 588 upregulated and 992 downregulated genes). Finally, in the P4 stage, the comparison between P1S–P4S and CK identified 4664 DEGs (with 2177 upregulated and 2487 downregulated genes) (Figure 5A). These results indicate that exposure to P1 MaE induced a substantial number of genes to exhibit differential expression. However, as conditions reverted to CK in P2 and P3, the number of differentially expressed genes gradually decreased, only to substantially increase again in P4 when re-exposed to MaE. The differential expression patterns across P1 to P4 included sustained upregulation, sustained downregulation, fluctuating expression, and transient expression (Figure 5B).

In the P1 (S vs. CK) comparison, various biological processes related to environmental and stress responses, such as the cellular response to high light intensity, hypoxia, salt stress, regulation of defense response, and defense response to bacterium, were significantly enriched. Additionally, processes associated with epigenetic gene expression regulation, histone H3-K9 dimethylation, chromatin remodeling, and developmental processes (e.g., root hair cell tip growth, meristem development, and plant-type primary cell wall biogenesis) were enriched (Figure 5C). In the P2 (P1S–P2CK vs. CK) comparison, there was enrichment in a wide range of biological processes related to both biotic and abiotic stress responses and light response, including processes like the response to oxidative stress, hypoxia, and light intensity. Cell wall formation processes, such as plant-type primary cell wall biogenesis and cell wall organization, were enriched, along with developmental processes such as development growth and root morphogenesis. Furthermore, other functions, including anaerobic respiration and spermine catabolic processes, were also enriched (Figure 5D). For the P3 (P1S–P3CK vs. CK) comparison, enrichment was observed in processes related to both biotic and abiotic stress responses, epigenetic gene regulation (e.g., epigenetic regulation of gene expression and DNA methylation within a CG sequence), and some developmental processes, while specific items related to root development and cell wall synthesis were not significantly enriched (Figure 5E). The GO enrichment results for P4 (P1S–P4S vs. CK) were largely similar to those in the P1 comparison, covering processes related to stress responses, epigenetic regulation, developmental processes (e.g., cell tip growth, the regulation of the developmental process), cell wall thickening (plant-type primary cell wall biogenesis), and various other processes potentially associated with stress resistance (e.g., the positive regulation of superoxide dismutase activity and wax biosynthetic process) (Figure 5F).

According to the identification procedure for transcriptional memory genes (TMGs), MaE-responsive genes with similar differential expression patterns in the intersections of S vs. CK, P1S–P2CK vs. CK, and P1S–P4S vs. CK were designated as TMG2. Furthermore, TMG3 was defined as the subset of TMG2 that shared similar differential expression patterns with P1S–P3CK in the intersection. Using this approach, a total of 399 downregulated TMG2 genes were identified, with 200 of them also belonging to TMG3 (Figure 6A). Additionally, 220 upregulated TMG2 genes were identified, with 51 of them being part of TMG3 (Figure 6B). The GO enrichment analysis results revealed that TMG2 genes were primarily involved in processes related to tissue development, root hair tip growth, response to light, response to oxidative stress, epigenetic processes, and photosystem stabilization (Figure 6C). On the other hand, TMG3 genes, being a subset of TMG2, encompassed genes associated with processes such as tissue development, the response to oxidative stress, epigenetic processes, and other related processes (Figure 6D). The information on TMG genes is listed in Appendix A.

### 2.5. Gene Co-Expression Modules Related to Measured Traits

The WGCNA analysis obtained a total of 21 network modules comprising 431,491 edges. The results of the module–trait association analysis revealed significant relationships between weaken traits and several modules (green, light green, magenta, purple, and turquoise) (PCC > 0.5, *p* < 0.05). Due to the similarity of weaken traits on the reaction norm, it is common for a module to be correlated with multiple weaken traits (and likewise for strengthen traits). For instance, the magenta module displayed significant associations with the number of leaves (PCC = 0.82, *p* < 0.01), total leaf area (PCC = 0.6, *p* = 0.02), content of Chl b (PCC = 0.56, *p* = 0.03), Fv/Fm (PCC = 0.85, *p* < 0.01), and Fm (PCC = 0.79, *p* < 0.01). The turquoise module, while only significantly associated with Fm (PCC = 0.64, *p* = 0.01), exhibited a near-significant correlation with other weaken traits, such as the total leaf area (PCC = 0.48, *p* = 0.07) and content of Chl b (PCC = 0.42, *p* = 0.1) (Figure 7A). Similarly, the blue, light cyan, red, salmon, and yellow modules displayed significant associations (PCC > 0.5, *p* < 0.05) with one or more strengthen traits. For example, the blue module showed significant correlations with SOD (PCC = 0.63, *p* = 0.01), POD (PCC = 0.63, *p* = 0.01), soluble sugar (PCC = 0.66, *p* < 0.01), and protein (PCC = 0.74, *p* < 0.01). The yellow module exhibited significant associations with SOD (PCC = 0.59, *p* = 0.02), proline (PCC = 0.54, *p* = 0.04), soluble sugar (PCC = 0.55, *p* = 0.03), and protein (PCC = 0.74, *p* < 0.01), with a near-significant correlation with POD (PCC = 0.49, *p* = 0.06) (Figure 7A). The functional analysis of genes within each module indicated that weaken trait-related modules harbored genes linked to photosynthesis and developmental growth (Figure 7B). This elucidates the connections between these modules and morphological traits, which depend on developmental processes, including the biomass, leaf count, and total leaf area. On the other hand, strengthen trait-related modules recruited genes associated with light response and oxidative stress response (Figure 7B). The blue and purple modules serve as transitional modules in the network topology, connecting the strengthen and weaken trait modules (Figure 7C). These modules exhibit shared functions involving genes related to developmental growth, photosynthesis, and light response. TMGs represent 13.16% of weaken trait-related modules, 24.90% of strengthen trait-related modules, and only 8.87% of other modules (Figure 7D), suggesting that modules associated with transgenerational plasticity tend to recruit a higher proportion of TMGs.

The top 10% of nodes by degree in each module were defined as hub genes. According to this criterion, a total of 359 hub genes were identified in both the weaken trait-related and strengthen trait-related modules. Categorized by the biological processes they participate in, 94 are associated with growth and development, 12 with epigenetic processes, 12 with the response to oxidative stress, 26 with photosynthesis, and 45 with the response to light stimulus. The weaken trait-related modules (turquoise, purple, magenta, light green, and green) contain 39 TMG2 and 52 TMG3, most of which are primarily associated with growth and development. The strengthen trait-related modules (yellow, salmon, red, light cyan, and blue) contain 18 TMG2 and 5 TMG3, most of which are primarily associated with functions related to the response to oxidative stress. Therefore, TMGs account for 32.59% of the hub genes. The information on hub genes is listed in Appendix A.

To examine the connection between TMGs and module functions, we identified gene nodes associated with developmental growth and response to oxidative stress, along with their immediate network neighbors, from both weaken trait-related and strengthen trait-related modules. The developmental growth-related subgroup included 376 gene nodes, comprising 73 TMG2 and 57 TMG3, as well as developmental growth-associated transcription factors like *AIL5*, *ARF*, *HAM3*, and *BLH2* (Figure 8A). Several genes within this subgroup exhibited expression patterns akin to those in weaken traits (Figure 8B). These genes were notably suppressed when exposed to MaE in the P1 stage, with this suppression persisting into P2 and gradually recovering in P3. Upon re-exposure to MaE in P4, the downregulation was less pronounced than in P1 and P2. Notably, TMGs related to leaf development, such as *MAPKKK7* and *BLH2*, were part of this subgroup. The submodule related to the response to oxidative stress comprised 149 gene nodes, featuring 8 TMG2 genes and 4 TMG3 genes. Additionally, this submodule included various superoxide dismutase genes, such as *FSD2*, *CSD3*, and *FSD3* (*FSD3* is TMG3). Furthermore, it encompassed other TMGs associated with anti-oxidative stress functions, including *GPX6*, *HSP17*, and *SIA1* (Figure 8C). Many genes within this submodule exhibited expression patterns similar to those observed in strengthen traits. They were activated and upregulated upon exposure to MaE in the P1 stage, gradually returning to CK levels in P2 and P3. However, in the P4 stage, upon re-exposure to MaE, they were reactivated and displayed higher expression levels than in the P1 stage (Figure 8D).

### 2.6. Validation of Gene Expression Revealed by RNA Sequencing

The expression patterns of eight TMG3 genes were validated using qRT-PCR to assess the accuracy of RNA sequencing results (Figure 9). Comparative analysis demonstrated a high degree of consistency between the qRT-PCR findings and the RNA sequencing results. For instance, FSD3 exhibited significant upregulation under high MaE concentration conditions in P1 (log2 fold change S vs. CK = 1.1 in RNA sequencing), and qRT-PCR results showed that the relative expression of *FSD3* in P1S–P3CK samples was approximately two times that of CK (Figure 9A). These outcomes affirm the accuracy and reliability of RNA sequencing results and also substantiate that TMG3 genes maintain the same differential expression patterns in the P3 stage despite the restoration of cultivation conditions to CK in both P2 and P3.

## 3. Discussion

### 3.1. TGP as a Mechanism for Offspring Resilience to MaE in Duckweed

Eutrophication and global warming can intensify cyanobacterial blooms, posing significant risks to aquatic ecosystems. In this study, duckweed exhibited a strong phenotypic plasticity in response to MaE, adapting its traits in a concentration-dependent manner. During the P1 stage, a biphasic dose–response curve emerged, with low MaE concentrations promoting and high concentrations inhibiting trait values related to biomass, leaf count, total leaf area, maximum fluorescence (Fm), the ratio of variable to maximum fluorescence (Fv/Fm), and chlorophyll b content (Figure 2). Similar growth patterns have been observed in other aquatic plants, such as *Potamogeton malaianus* and *Ceratophyllum oryzetorum*, which can thrive under low MaE concentrations [16,29], suggesting that this ‘low promotion and high inhibition’ phenomenon may occur naturally. Many plants exhibit the phenomenon known as hormesis, where low doses of toxins stimulate growth and high doses inhibit it [30,31]. This may be due to the activation of moderate defense mechanisms that eliminate damage and simultaneously increase photosynthesis and dark respiration efficiency. This leads to a positive energy balance, where energy assimilation exceeds dissimilation, thereby stimulating plant growth and productivity [31].

Like microcystins, MaE has the potential to induce oxidative stress, leading to lipid membrane peroxidation, genotoxicity, and the regulation of apoptosis [32]. The increased root length and activities of antioxidative enzymes (SOD, POD, CAT, and APX) in response to MaE suggest a strategic adaptation to enhance nutrient uptake and mitigate oxidative stress induced by MaE [18,33].

Several observed plastic trait variations are present in the asexual progeny of the P2 generation (Figure 3), indicating a significant manifestation of TGP. The doubling time for duckweed biomass under optimal conditions ranges from 1.34 to 4.54 days [25,34]. Therefore, each stage of our study encompasses at least three generations of asexual reproduction, suggesting that TGP for these traits occurs across multiple generations. Both weaken traits, which are linked to leaf growth and photosynthesis, and strengthen traits, associated with antioxidant resistance and adaptive physiological changes, demonstrate the occurrence of TGP. However, their impacts on the individual fitness of duckweed under MaE stress may differ. Weaken traits help alleviate the inhibitory effects of MaE on growth and development, while strengthen traits enhance resistance to MaE stress. Despite these differences, both trait categories exhibit regulatory responses aimed at improving fitness. As a result, the exposure of duckweed parents to MaE may lead to more stable fitness in their offspring under stressful conditions, exemplifying the ‘jack of all trades’ strategy. This strategy allows adaptive plasticity to help maintain fitness stability in adverse environments [35].

Empirical evidence supports the adaptability of TGP when the environments of parents and offspring are similar [6,36,37]. In nutrient-rich or contaminated water bodies, cyanobacterial blooms frequently recur, increasing the likelihood of alignment between parental and offspring environments. Field observations indicate that Lemna species often coexist with toxic cyanobacterial blooms [18], and duckweed collected from these environments has demonstrated enhanced tolerance to Microcystis toxicity [17]. This suggests that TGP plays a crucial role in improving duckweed’s resistance to harmful cyanobacterial blooms.

### 3.2. Linking Gene Expression to Adaptive Phenotypic Changes

Given duckweed’s clonal reproduction, the observed phenotypic variations are likely due to modulation of gene expression rather than genetic differences. RNA sequencing analyses reveal a strong correlation between trait variations and gene expression patterns in duckweed. The number of DEGs peaks during the P1 and P4 stages of MaE exposure but gradually decreases as the growth environment returns to CK conditions in the P2 and P3 stages (Figure 5A). This temporal trend parallels the shifts seen in multiple traits. Additionally, certain DEGs are functionally linked to these phenotypic changes, such as the significant enrichment of genes related to root development in P1, which corresponds with the notable elongation of roots observed under high MaE exposure. It is important to recognize that phenotypic plasticity can manifest in various forms, including morphological, physiological, and gene expression plasticity. The ultimate phenotype of an organism results from the integration of plasticity across these different levels [38]. The relationship between gene expression and phenotypes exemplifies this complex integration.

Research into the molecular basis of phenotypic plasticity suggests the presence of key genes, often termed ‘genetic switch genes’, that serve as central components within gene networks. These genes can sense environmental cues and regulate alternative phenotypes, significantly influencing the overall plasticity and robustness of organisms [39,40]. Through gene co-expression analysis, we identified gene modules associated with weaken traits and strengthen traits. Weaken traits require developmental switches to express trait variations. Gene modules linked to weaken traits include numerous TMGs associated with developmental processes, many of which function as hub genes (Figure 8A), highlighting their critical role in the TGP of weaken traits. For example, *MAPKKK7* (novel.1153, TMG2) and *BLH2* (Lminor_018634, TMG3 and TF) have been shown to influence cell expansion and leaf morphogenesis [41,42], respectively, and are included in the weaken trait modules (Figure 8A). In contrast, strengthen traits are primarily associated with responses to oxidative stress. Although the oxidative stress-responsive submodule contains fewer TMGs, some of these genes are functionally linked to antioxidative mechanisms, including superoxide dismutase *FSD3* (Lminor_014667, TMG3), *GPX6* (Lminor_015761, TMG2), *HSP17* (Lminor_002987, TMG2), and *SIA1* (Lminor_015291, TMG2) [43,44,45]. These genes, incorporated into strengthen trait modules, exhibit expression patterns in the P1–P4 stages that align with those of related traits (Figure 8C,D), suggesting their essential roles in the TGP of duckweed.

## 4. Material and Methods

### 4.1. Cultivation of M. aeruginosa and Duckweed

The *M. aeruginosa* strain (FACHB-905), originally from Dianchi Lake, was obtained from the Freshwater Algae Culture Collection of the Institute of Hydrobiology (FACHB-Collection) at the Chinese Academy of Sciences. *M. aeruginosa* was cultured in a modified HGZ-145 nutrient solution [46], which was sterilized and placed into 500 mL flasks. These flasks were kept in a climate-controlled chamber set to 24–26 °C with a 12:12 light–dark cycle. To ensure the algal cells remained suspended, the flasks were agitated three times daily. The duckweed strain used in our study was *Lemna minor* 1084. This strain originates from the natural population of duckweed in Zhongdian County, Yunnan Province, and was identified, preserved, and provided by the Chengdu Institute of Biology, Chinese Academy of Sciences, ClB, CAS. Barcoding was employed to further verify the species identity of the plant material, following the methodologies described in previous studies [47,48]. In brief, DNA was extracted using the FastPure Plant DNA Isolation Mini Kit (Vazyme, Nanjing, China), and plastid intergenic spacers were amplified with the *atpF*-*atpH* primers (forward: 5′-ACTCGCACACACTCCCTTTCC-3′, reverse: 5′-GCTTTTATGGAAGCTTTAACAAT-3′) and *psbK*-*psbI* primers (forward: 5′-TTAGCATTTGTTTGGCAAG-3′, reverse: 5′-AAAGTTTGAGAGTAAGCAT-3′). Sanger sequencing was performed, and the sequences were subjected to blastn alignment. The results confirmed that the obtained sequences were most closely related to *L. minor* (Appendix A). The strain was clonally propagated for genetic consistency and cultured in HGZ-145 nutrient solution [46] in a climate-controlled chamber at 24–26 °C, with light intensity of 2200–2500 lux (approximately 41–46 µmol m^−2^ s^−1^) and a 16:8 light–dark cycle. After eight weeks, healthy and uniformly grown duckweed individuals were selected for the experiments.

### 4.2. Treatment of Duckweed Exposure to MaE

Cyanobacterial bloom densities in Lake Dianchi can reach levels between 4 × 10^5^ cells/mL and 1.3 × 10^7^ cells/mL during bloom periods [16,49]. Accordingly, four treatment gradients were established for Microcystis aeruginosa: a control group (CK, 0 cells/mL), low (4 × 10^4^ cells/mL), medium (4 × 10^5^ cells/mL), and high (S, 4 × 10^6^ cells/mL) (Figure 1A). Cultures of *M. aeruginosa* were initiated at 2 × 10^6^ cells/mL and allowed to grow to 4 × 10^6^ cells/mL over six days before being diluted to lower concentrations. Daily cell counts were performed to maintain consistent density, and MaE was harvested via centrifugation at 6000 rpm for 10 min for the following exposure experiment. Treatments were refreshed weekly.

To assess the impact of parental exposure to *M. aeruginosa* on offspring in duckweed, each gradient, consisting of 16 duckweeds, was subjected to an exposure experiment (i.e., stress by MaE) in controlled growth chambers maintained at 25 °C with a 16 h light and 8 h dark cycle for ten days (P1). During this period, the duckweed was capable of asexual reproduction through cloning (Figure 1A). Given that the results indicated the most pronounced phenotypic effects in the plants treated with high concentrations (S), and our field surveys have shown that S conditions frequently occur under natural conditions, we therefore employed the progeny from the high-concentration treatment group for subsequent experiments. Following P1, the offspring of CK and S were continuously cultivated under CK conditions for two additional growth cycles under control conditions (P2 and P3, Figure 1B). Specifically, 16 duckweed plants were selected from P1-CK and P1-S and transferred to CK conditions for ten days (P2). This process was repeated for a third growth cycle (P3). At the end of P3, 16 offspring from both the P1-CK and P1-S parental groups were chosen and cultivated under CK and S conditions (P4) to evaluate the potential effects of parental exposure to *M. aeruginosa* on their fitness (Figure 1B).

### 4.3. Trait Measurement

The following traits were measured for duckweed individuals across growth periods P1 to P4, with an average of six replications for each measurement. (1) Root length: the length of the roots was calculated from photographs using ImageJ version 1.53v. (2) Biomass: fresh and dry weights of duckweed were used to determine biomass. (3) Number of leaflets: the total number of leaflets was counted at the end of each growth period. (4) Leaf area: the total leaflet area was measured from photographs using ImageJ. (5) Photosynthetic parameters: baseline fluorescence (F0), maximum fluorescence (Fm), and the ratio of variable to maximum fluorescence (Fv/Fm) were measured using a modulating pulse amplitude modulation (PAM) fluorimeter (Junior PAM, Heinz Walz GmbH, Germany). (6) Chlorophyll content: the contents of chlorophyll a and chlorophyll b were determined using a spectrophotometric method. (7) Antioxidant enzyme activity: the activities of superoxide dismutase (SOD), peroxidase (POD), catalase (CAT), and ascorbate peroxidase (APX) were measured using a Lambda12 UV-spectrophotometer (Perkin Elmer Instrument Co., Ltd., Waltham, MA, USA) through UV spectrophotometry. (8) Soluble compounds: the contents of proline, soluble sugars, and soluble proteins were measured using spectrophotometric methods.

All physiological trait measurements were conducted according to the manufacturers’ instructions. Fisher’s least significant difference (LSD) test was employed to assess significant differences between treatments.

### 4.4. Transcriptome Sequencing and Data Analysis

Based on the results of trait measurements, transgenerational plasticity in duckweed was most pronounced under high-concentration MaE conditions (S). Therefore, this treatment group was selected for RNA sequencing. Five sample groups were subjected to sequencing: the control (CK) and high-concentration (S) groups from the P1 generation, the offspring of the S group in the P2 and P3 period under control conditions (i.e., P1S–P2CK and P1S–P3CK), and the descendants of the S group re-exposed to high-concentration MaE in the P4 period (P1S–P4S). Leaflet samples were collected after each cultivation period, with each sample group consisting of three biological replicates. Total RNA was extracted from the leaflet samples using RNA iso reagent (TaKaRa, Dalian, China), following the manufacturer’s protocol. RNA concentration was quantified using a Nanodrop 2000 (Thermo Scientific, Waltham, MA, USA), and RNA integrity was assessed using an Agilent Bioanalyzer 2100 (Agilent Technologies, Santa Clara, CA, USA). High-quality RNAs were then used for RNA sequencing via Novogene Bioinformatics Technology Co., Ltd. (Beijing, China). The library construction process involved fragmentation of purified mRNA, first- and second-strand cDNA synthesis, adaptor ligation, PCR, and quality assessment and clustering. Finally, the cDNA libraries were subjected to 150 bp paired-end sequencing on the Illumina NovaSeq 6000 System.

High-quality clean data were obtained by eliminating reads containing adapters, poly-N sequences, and low-quality reads from the raw sequencing data. Paired-end clean reads were aligned to the duckweed genome [50] using Hisat2 v2.0.5 with default parameters. New transcript assembly was performed using StringTie software (version 1.3.3b). The resulting transcripts were further annotated against the TAIR10 database via BLAST in addition to the existing genome annotation. The featureCounts tool in the Subread software (version 2.0.3) was employed to count the number of reads mapped to each gene, allowing for the calculation of FPKM values. DEGs were identified using the DESeq2 R package, with CK samples as the control and significance set at an adjusted *p*-value < 0.05 and log2 fold change > 0.5 [11].

### 4.5. Identification of Transcriptional Memory Genes

TMGs were identified through a three-step process [11], focusing on consistent expression changes across exposures to MaE and subsequent non-exposure generations. To elaborate, genes with the same differential expression patterns in the P1 stage between S and CK and in the P4 stage between P1S–P4S and CK were identified, resulting in robust MAE-responsive genes. Then, the intersection of robust MAE-responsive genes with P1S–P2CK was taken to obtain genes whose expression patterns continued into P2 (TMG2). Finally, the intersection of TMG2 with P1S–P3CK was taken to obtain genes whose expression patterns could continue into P3 (TMG3). Gene Ontology (GO) enrichment analysis of MaE-responsive genes and TMGs was performed using the R package TopGO (version 2.58.0.), with GO terms having a corrected *p*-value of ≤0.05 considered significantly enriched.

### 4.6. Gene Co-Expression Network Analysis

A co-expression network was constructed using the R package WGCNA [51]. Initially, the FPKM values of differentially expressed genes were standardized using the limma R package. Transcripts with missing entries were then filtered out using the goodSamplesGenes function. Next, the scale-free topology for various soft thresholding powers was calculated using the pickSoftThreshold function. Based on the results, the power parameter was set to 20 to construct the network (Appendix A). Other parameters were set as follows: minModuleSize = 30 and mergeCutHeight = 0.25.

The eigengenes of each module were calculated using the moduleEigengenes function, representing the weights of the first principal component for the gene expression matrix of each module. The Pearson correlation coefficient (PCC) was then computed between the trait values and the eigengenes for each sample to identify modules correlated with the traits of interest. Modules meeting the criteria of PCC > 0.5 and *p* < 0.05 were considered associated with the trait. Due to the large number of edges generated, subsequent analyses focused on the top 30% of edges ranked by weight. The top 10% of nodes in each module, defined by their degree, were designated as hub genes for that module.

### 4.7. Quantitative Real-Time PCR Verification of Genes

To validate the expression patterns identified through RNA sequencing, we selected several key TMG3 genes for Quantitative Real-Time PCR (qRT-PCR) verification. This selection included one iron superoxide dismutase gene (FSD3) and seven transcription factors (TFs) from the MYB, NAC, WOX, bHLH, bZIP, Nin-like, and TALE families. A total of 1.0 µg of total RNA from each sample was reverse-transcribed into cDNA using a PrimeScriptTM RT reagent kit (TaKaRa, Dalian, China). The qRT-PCR experiments were conducted using TB Green™ Premix Ex Taq™ II (TaKaRa, Dalian, China) on the Roche LightCycler 96 system (Roche, Penzberg, Germany), following the manufacturer’s instructions. *UBQ10-2* was selected as the internal reference gene, and each reaction was performed in triplicate. Relative expression changes among different treatments were quantified using the 2^−ΔΔCt^ method, with primer details provided in Appendix A.

## 5. Conclusions

In conclusion, duckweed exhibits strong phenotypic plasticity in response to MaE. Traits such as biomass, leaf count, total leaf area, Fm, Fv/Fm, and chlorophyll b content showed enhancement at low MaE concentrations but were suppressed at high concentrations in parental generations. In contrast, the root length, activities of antioxidative enzymes, proline content, soluble sugar, and protein content significantly increased with higher MaE concentrations, contributing to duckweed’s tolerance to elevated MaE levels. The persistence of these trait variations across generations highlights a robust transgenerational plasticity (TGP), which enhances the fitness of descendants upon re-exposure to MaE. Notably, 619 genes exhibit transcriptional memory, maintaining expression patterns similar to those of their MaE-exposed parents even under MaE-free conditions. Some of these genes are associated with the observed trait variations during the TGP process. These findings underscore the critical role of TGP in duckweed’s adaptation to cyanobacterial blooms, suggesting its contribution to the adaptive evolution of floating plants in challenging aquatic environments.

## Figures and Tables

**Figure 1 ijms-25-13027-f001:**
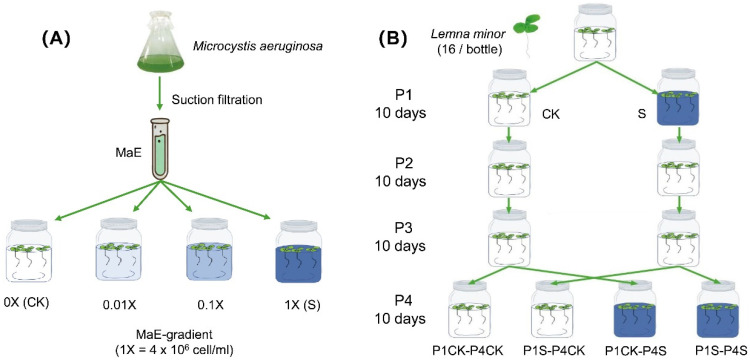
Experimental design. (**A**) Acquisition of different MaE concentrations. CK (0 cells/mL), low (4 × 10^4^ cells/mL), medium (4 × 10^5^ cells/mL), and S (4 × 10^6^ cells/mL). (**B**) Duckweed exposure to various MaE concentrations in P1, followed by two additional cultivation cycles in P2 and P3. In the P4 stage, descendants originating from P1-CK and P1-S were cultivated separately under CK and S conditions.

**Figure 2 ijms-25-13027-f002:**
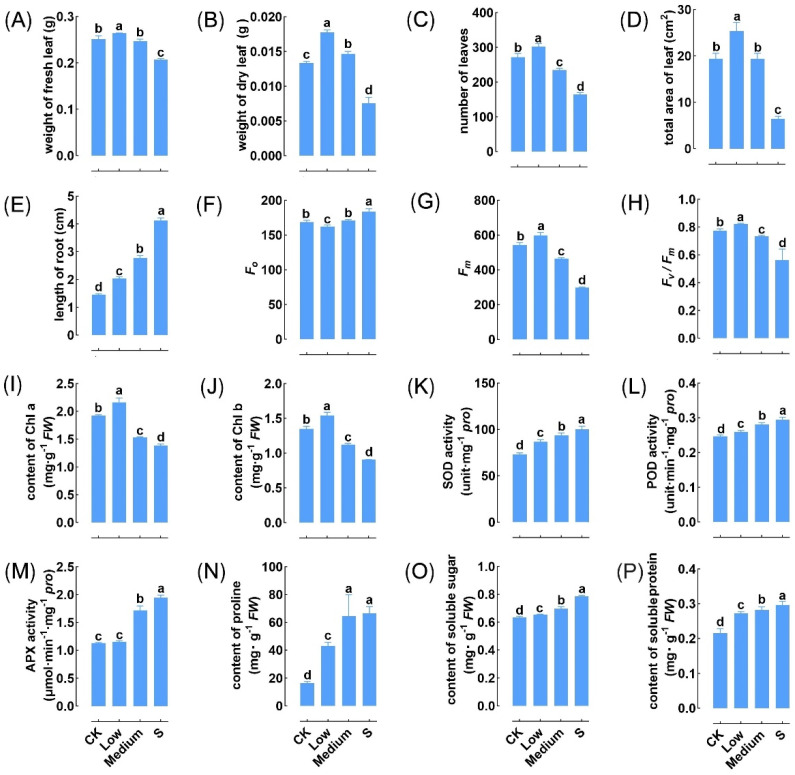
Variation of duckweed traits in response to different concentrations of MaE in P1. (**A**,**B**) Biomass of fresh leaf (**A**) and dry leaf (**B**). (**C**) Number of leaves. (**D**) Total area of leaf. (**E**) Length of root. (**F**) F0. (**G**) Fm. (**H**) Fv/Fm. (**I**,**J**) Content of Chl a (**I**) and Chl b (**J**). (**K**–**M**) Activity of SOD (**K**), POD (**L**), and APX (**M**). (**N**) Content of proline. (**O**) Content of soluble sugar. (**P**) Content of soluble protein. Different letters indicate significant differences among groups determined using Fisher’s LSD test.

**Figure 3 ijms-25-13027-f003:**
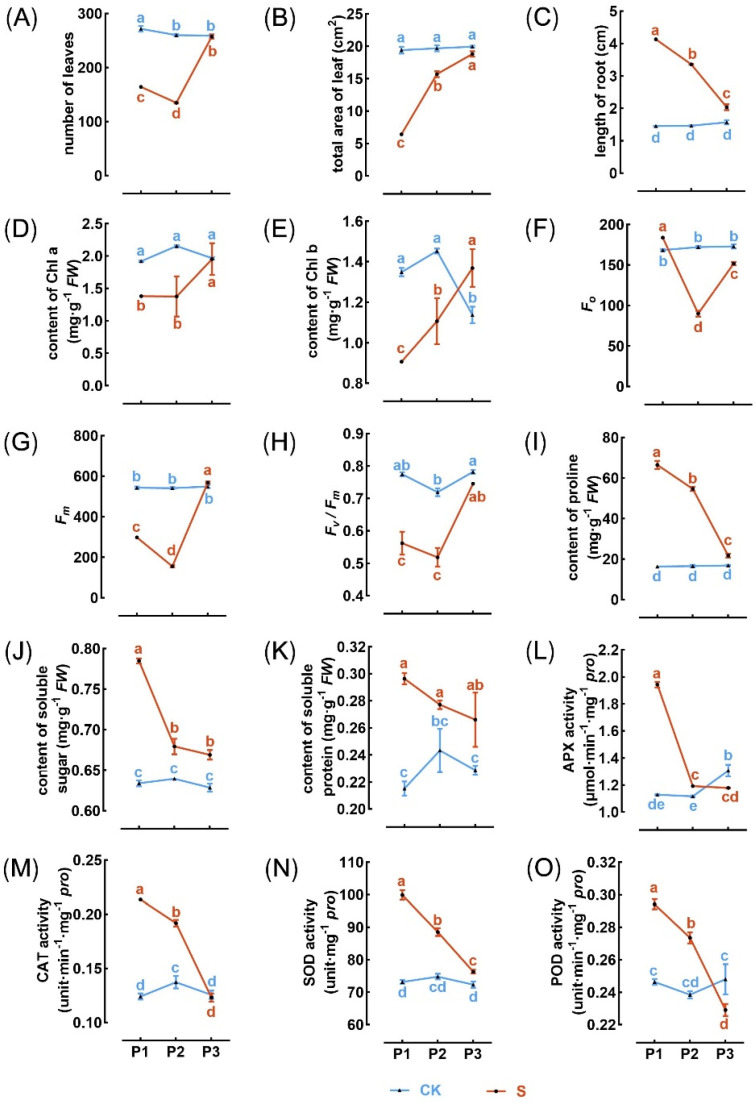
The trait variation in P1-P3 stage. (**A**) Number of leaves. (**B**) Total area of leaf. (**C**) Length of root. (**D**,**E**) Content of Chl a (**D**) and Chl b (**E**). (**F**) F0. (**G**) Fm. (**H**) Fv/Fm. (**I**,**K**) Content of proline (**I**), soluble sugar (**J**), and soluble protein (**K**). (**L**–**O**) The activity of antioxidant enzymes, including APX (**L**), CAT (**M**), SOD (**N**), and POD (**O**). The red line and blue line represent the parent environments of P1-S and P1-CK, respectively. Different letters indicate significant differences among groups determined using Fisher’s LSD test.

**Figure 4 ijms-25-13027-f004:**
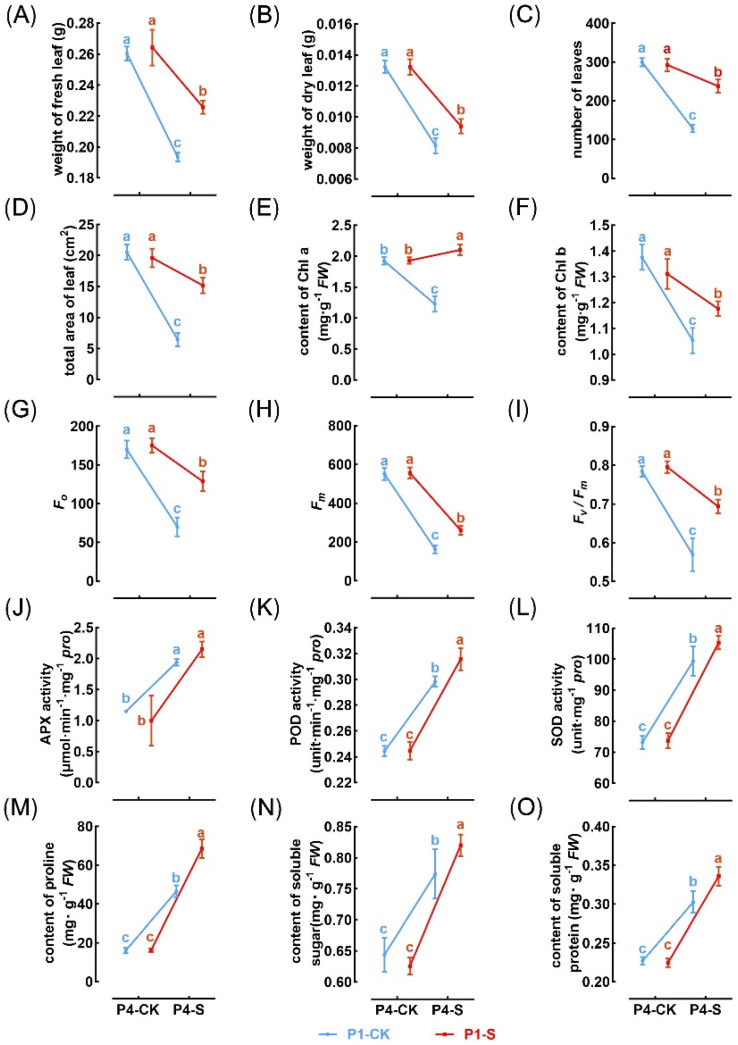
Reaction norm of trait expression in P4 individuals exposed to CK (0 cells/mL) and S (4 × 10^6^ cells/mL). The blue and red lines indicate the offspring of P1-CK and P1-S, respectively. (**A**,**B**) Biomass of fresh leaf (**A**) and dry leaf (**B**). (**C**) Number of leaves. (**D**) Total area of leaf. (**E**,**F**) Content of Chl a (**E**) and Chl b (**F**). (**G**) F0. (**H**) Fm. (**I**) Fv/Fm. (**J**–**L**) The activity of antioxidant enzymes, including APX (**J**), POD (**K**), and SOD (**L**). (**M**–**O**) Content of proline (**M**), soluble sugar (**N**), and soluble protein (**O**). Different letters indicate significant differences among groups determined using Fisher’s LSD test.

**Figure 5 ijms-25-13027-f005:**
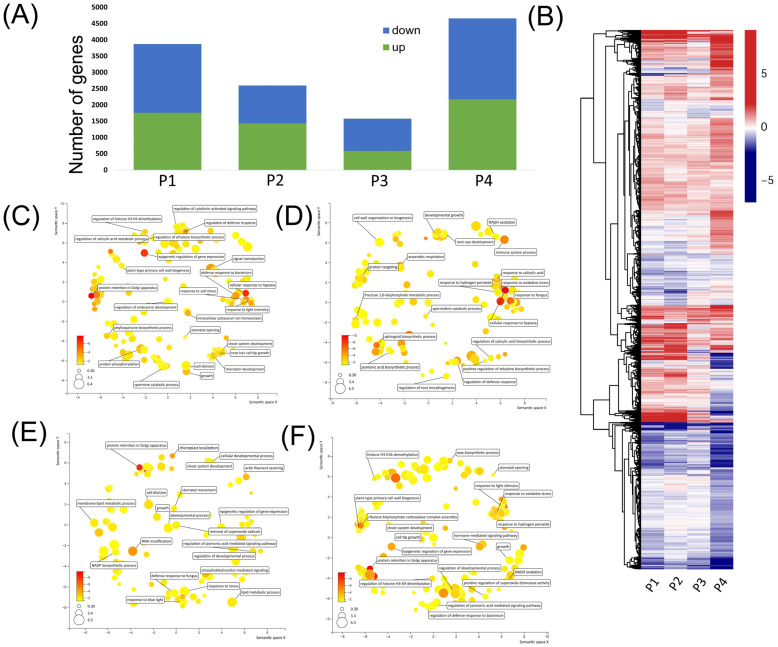
Expression and function of differentially expressed genes relative to CK in each period. (**A**) Number of upregulated and downregulated genes in P1 (S vs. CK), P2 (P1S–P2CK vs. CK), P3 (P1S–P3CK vs. CK), and P4 (P1S–P4S vs. CK). (**B**) Heatmap of differentially expressed genes. The color represents log2 fold change values; negative values (blue) represent downregulation, and positive values (red) represent upregulation. (**C**–**F**) Biological processes of differentially expressed genes of P1 (**C**), P2 (**D**), P3 (**E**), and P4 (**F**). The color and size were adjusted to *p*-value and log10 *p*-value, respectively.

**Figure 6 ijms-25-13027-f006:**
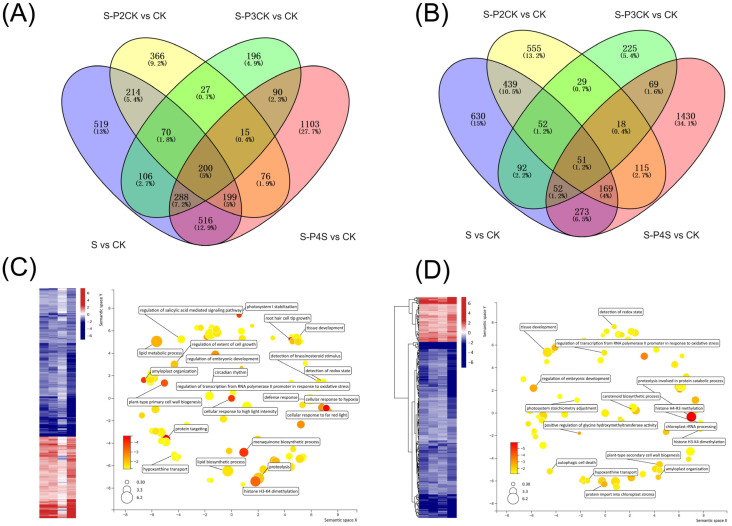
Identification and characterization of MaE-responsive TMGs. (**A**,**B**) Identification of downregulated (**A**) and upregulated (**B**) TMGs. The intersection of S vs. CK, P1S–P4S vs. CK, and P1S–P2CK vs. CK was identified as TMG2. The intersection of TMG2 and P1S–P4S vs. CK was identified as TMG3. (**C**,**D**) Expression and biological processes of TMG2 (**C**) and TMG3 (**D**). For the heatmap, color was adjusted to log2 fold change of comparison pairs. For GO map results, the color and size were adjusted to *p*-value and log10 *p*-value, respectively.

**Figure 7 ijms-25-13027-f007:**
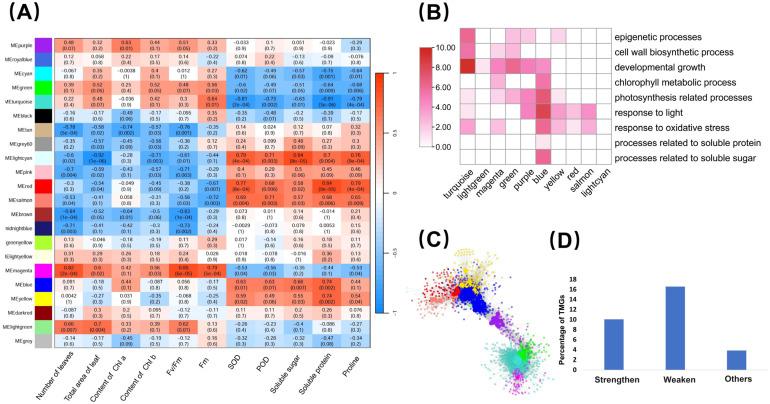
Identification and characterization of traits related gene co-expression modules. (**A**) Module–trait associations. (**B**) The number of genes involved in different biological processes within each trait-related module. The color intensity represents the magnitude of log2–transformed values. (**C**) A subnetwork composed of weaken trait-related modules (turquoise, light green, magenta, green, and purple) and strengthen trait-related modules (blue, yellow, red, salmon, and light cyan). (**D**) More TMGs were found in strengthen and weaken trait-related modules than in other modules.

**Figure 8 ijms-25-13027-f008:**
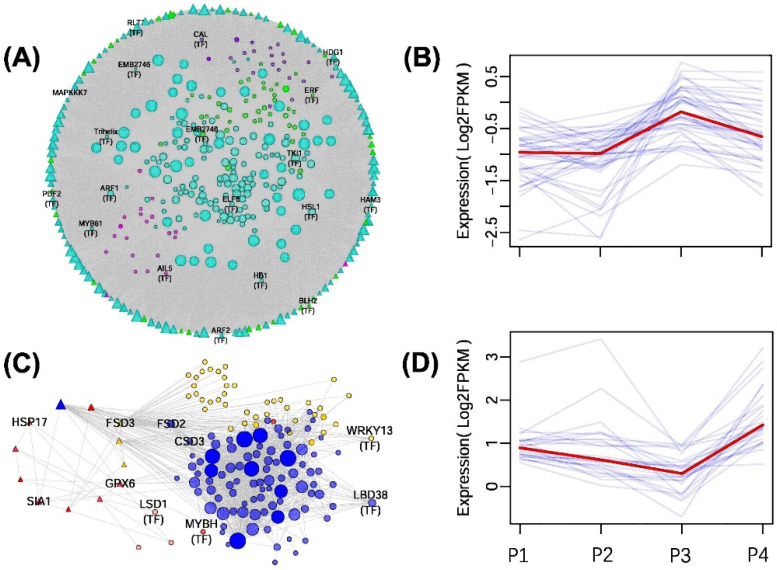
Subnetworks composed of gene nodes related to TGP traits and their immediate network neighbors. (**A**,**B**) Network of developmental growth-related genes (**A**). Some developmental growth-related genes exhibit a similar expression pattern to weaken traits, which means they are initially suppressed by MaE at the P1 stage, gradually recover to normal expression levels in the P2–P3 stages, and are less suppressed at the P4 stage when re-exposed to MaE compared to the P1 stage (**B**). (**C**,**D**) Network of oxidative responsive genes (**C**). Among them, some exhibit a similar expression trend to SOD, where they are induced by MaE at the P1 stage, gradually recover during the P2–P3 stages, and are upregulated to a level higher than that at the P1 stage when re-exposed to MaE at the P4 stage (**D**). For networks, transcriptional factors and some key genes were marked. TMGs are represented by triangle, and modules that the gene belongs to are represented by different colors. Node size is positively correlated with the degree of connectivity. In panels B and D, the blue lines represent the expression trends of individual genes, while the red lines represent the overall expression trend.

**Figure 9 ijms-25-13027-f009:**
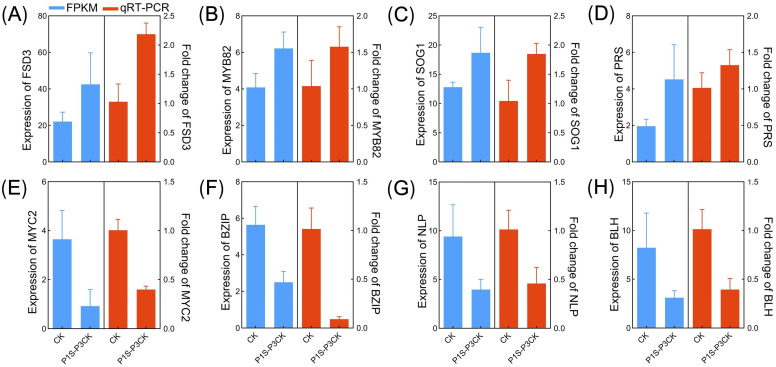
(**A**–**H**) Validation of expression pattern of eight TMGs via qRT-PCR. The blue and red bars represent the FPKM values and relative expression values from RNA-seq and the qRT-PCR results, respectively. For the qRT-PCR results, relative expression quantification was carried out using the 2^−ΔΔCt^ method, with *UBQ10-2* as the internal reference gene. Error bars represent the mean ± SE (n = 3).

## Data Availability

The data presented in this study are deposited in the SRA database of the NCBI repository, accession numbers SRR2648417-SRR2648431.

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
