# Peer review of "Transgenerational Plasticity Enhances the Tolerance of Duckweed (Lemna minor) to Stress from Exudates of Microcystis aeruginosa"

_ijms, 2024, doi:10.3390/ijms252313027_

Round 1
Reviewer 1 Report
Comments and Suggestions for Authors
Although this is a quite interesting manuscript, both in view of the methods used and the results obtained, I do not want to review a manuscript without identification of the biological object – we are in biology! The authors mentioned that “Duckweed, also sourced from the same institute” and in the headline that it is Lemna minor. This is not the way to identify duckweeds that presently consist of 35 species and two hybrids. There is no way to identify L. minor on the basis of morphology. State of art is using one or two of the plastidic markers, which means PCR and sequencing of the PCR fragment (cf. papers of M. Bog), plus applying the method of tubulin-based polymorphism (cf. papers of L. Morello). Moreover, the authors did not bother to mention the medium in which they cultivated the duckweed, the temperature or the light intensity.
After the authors provided these data I suggest to re-consider the manuscript for review.
Author Response
Comment 1: Although this is a quite interesting manuscript, both in view of the methods used and the results obtained, I do not want to review a manuscript without identification of the biological object – we are in biology! The authors mentioned that “Duckweed, also sourced from the same institute” and in the headline that it is Lemna minor. This is not the way to identify duckweeds that presently consist of 35 species and two hybrids. There is no way to identify L. minor on the basis of morphology. State of art is using one or two of the plastidic markers, which means PCR and sequencing of the PCR fragment (cf. papers of M. Bog), plus applying the method of tubulin-based polymorphism (cf. papers of L. Morello). Moreover, the authors did not bother to mention the medium in which they cultivated the duckweed, the temperature or the light intensity.
After the authors provided these data I suggest to re-consider the manuscript for review.
Response 1:
Thank you very much for your valuable comments. We fully agree with your opinion regarding the accurate identification of duckweed strain in biological experiments. We have revised our manuscript and provided detailed information for Lemna minor, Microcystis aeruginosa, and the experimental conditions in this study.
The Lemna minor material used in our study was not collected directly from the wild, but was purchased from the Chengdu Institute of Biology, Chinese Academy of Sciences. This institute is a highly renowned research institution recognized for the genetic identification and authenticity of its biological collections. Upon inquiry, the strain of the experimental material is Lemna minor 1084. Currently, we are conducting DNA barcode species identification on the duckweed used in our experiments. However, the editor has requested that the response to the reviewers be uploaded within ten days, so we are currently unable to provide this result. But we promise to provide it as soon as it is available.
The sentence in the ''method'' part was revised to provide more clarity on the source and cultivation conditions of the Lemna minor strain: '' The duckweed strain used in our study was Lemna minor 1084. This strain originates from the natural population of duckweed in Zhongdian County, Yunnan Province, and was identified, preserved, and provided by the Chengdu Institute of Biology, Chinese Academy of Sciences, ClB, CAS. The strain was clonally reproduced to ensure genetic consistency and was cultivated in HGZ-145 nutrient solution within a climate-controlled chamber, maintained at 24-26°C with a 16:8 light-dark cycle. ''
Reviewer 2 Report
Comments and Suggestions for Authors
In this manuscript, the authors investigated the transgenerational plasticity of duckweed and its adaptive role under stress from MaE during the bloom-forming process. The results indicated that the plastic changes can be retained through asexual reproductive cycles. Progeny from MaE-exposed lineages demonstrated enhanced fitness when re-exposed to MaE. Furthermore, 619 genes involved in maintaining the transcriptional memory were identified through transcriptome sequencing. The study is important to understand the potential mechanism of duckweed to tolerate MaE stress and provides new approaches for utilizing duckweed’s TGP in the bioremediation of detrimental algal blooms. However, the manuscript could not be accepted at its present form. I think that some points have to be addressed before acceptance.
1.In Figure 2, duckweed’s traits like weight of leaf, number of leaves exhibit “low promotion and high inhibition” phenomenon. Although in discussion part, the author mentioned that the similar growth patterns were observed in other aquatic plants, however, the potential internal mechanisms need to be discussed more deeply by searching for the newest related literature.
2.The article mentioned that Microcystis aeruginosa can produce microsystin and MaE, which can induce oxidative stress, and are toxic to posing significant threats to biosecurity, food webs, and public health. I suggest the author to determine the concentration of the above chemical material in low, medium and high concentrations treatments of MaE in P1, P2, P3 and P4 individuals, to reveal the relationship between antioxidative enzymes and MaE.
3. By RNA sequencing and WGCNA analysis, the key gene nodes related with the developmental growth were identified. However, the candidate genes seemed to be too numerous. The key genes related with TGP have not be focused. In the above trait variation analysis part, the content of Chl a, Chl b, F0, Fm, Fv/Fm, proline, soluble sugar, soluble protein, the activity of antioxidant enzymes, including APX, CAT , SOD , and POD have be demonstrated having close association with TGP for offspring resilience to MaE in duckweed. I suggest the authors to focus on the genes involved the synthesis or metabolic cycles of the above genes, to make the study having more coherence.
4. 8 genes are not enough for validation of gene expression revealed by RNA sequencing. The expression pattern of CK were all 1.0 level from Figure 9 (A to H). The expression pattern of CK can be removed. The expression patterns of all selected TMG3 genes from transcriptome and qRT-PCR can be put in one figure, to compare their expression trend .
5. The resolution of figure 5 (C to F) and figure 6 (C and D) are very low, which are fuzzy. The dpi of figures should be above 300.
Author Response
Comment 1: In Figure 2, duckweed’s traits like weight of leaf, number of leaves exhibit “low promotion and high inhibition” phenomenon. Although in discussion part, the author mentioned that the similar growth patterns were observed in other aquatic plants, however, the potential internal mechanisms need to be discussed more deeply by searching for the newest related literature.
Response 1: Thank you for your meticulous review of our manuscript and the valuable suggestions you have offered. The "low promotion and high inhibition" phenomenon caused by low concentrations of toxins is also common in other plants facing a variety of toxic substances and is referred to as hormesis. We have reviewed the most recent literature in this area of research and, based on the content of the literature, explained the possible mechanisms in our article. The following sentences have been incorporated into the corresponding section of the discussion to elucidate the "low promotion and high inhibition" phenomenon:
''Many plants exhibit the phenomenon known as hormesis, where low doses of toxins stimulate growth and high doses inhibit it [30,31]. This may be due to the activation of moderate defense mechanisms that eliminate damage and simultaneously increase photosynthesis and dark respiration efficiency. This leads to a positive energy balance, where energy assimilation exceeds dissimilation, thereby stimulating plant growth and productivity [31]. ''
Accordingly, two additional references have been added:
- Agathokleous, E.; Calabrese, E.J.; Barceló, D. Environmental Hormesis: New Developments. Sci. Total Environ. 2024, 906, 167450, doi:https://doi.org/10.1016/j.scitotenv.2023.167450.
- Erofeeva, E.A. Plant Hormesis: The Energy Aspect of Low and High-Dose Stresses. Plant Stress 2024, 100628, doi:https://doi.org/10.1016/j.stress.2024.100628.
Comment 2: The article mentioned that Microcystis aeruginosa can produce microsystin and MaE, which can induce oxidative stress, and are toxic to posing significant threats to biosecurity, food webs, and public health. I suggest the author to determine the concentration of the above chemical material in low, medium and high concentrations treatments of MaE in P1, P2, P3 and P4 individuals, to reveal the relationship between antioxidative enzymes and MaE.
Response 2: Thank you for your suggestions. As we stated in our article, MaE and microcystin are two different toxins released by Microcystis aeruginosa during different stages of its growth. MaE is primarily produced during the logarithmic growth phase of M. aeruginosa, while microcystin is typically released into the environment after cell death during algal blooms. These two toxins differ significantly in their stages of action and chemical composition. In this study, we focus on MaE to explore its impact on duckweed and the role of transgenerational plasticity in duckweed's adaptation to MaE.
Since MaE is a mixture, it is difficult to directly define its concentration; therefore, we use the number of M. aeruginosa cells per milliliter of culture medium to indirectly reflect the concentration of MaE. The cell count for each concentration gradient was set based on the results of our field surveys, as well as the description of cyanobacterial cell count gradients in the WHO (2003) guidelines for Safe Recreational Water Environments. In the experimental design, we only applied treatments with low, medium, and high concentrations during the P1 phase. The results demonstrated that the most pronounced phenotypic plasticity response occurred under high concentration conditions, and high concentrations are more reflective of the actual concentrations encountered in natural environments. Consequently, we selected the progeny from the high concentration treatment for further research. Under P2-P3 conditions, we cultivated the clonal offspring from the high concentration treatment in control (CK) environmental condition, Then high concentration treatments were reapplied again to assess changes in the fitness of the progeny in P4 stage. Therefore, the MaE concentration for P2 and P3 was set to 0, while the MaE treatment concentration for P4 was equivalent to that of the high concentration. Your suggestion to determine the concentration of MaE at low, medium, and high concentrations and its relationship with antioxidant enzymes is insightful for our future research. We will consider this in our subsequent studies to reveal the dynamic changes of both toxins during the occurrence of blooms and their impact on duckweed and other aquatic plants.
To elucidate our research context and experimental design more clearly, we have made the following revisions to the manuscript:
''Numerous studies have focused on the deleterious effects of microcystin on aquatic organisms. However, it is crucial to recognize that during the proliferation of M. aeruginosa, a highly toxic exudate mixture, henceforth referred to as MaE, is also secreted. This complex of compounds can influence other aquatic organisms at an earlier stage, posing significant threats to biosecurity, food webs, and public health even before the bloom explosion [17–19]'' (In introduction)
''Given that the results indicated the most pronounced phenotypic effects in the plants treated with high concentrations (S), and our field surveys have shown that S condition frequently occur under natural conditions, we therefore employed the progeny from the high concentration treatment group for subsequent experiments. Following P1, the offspring of CK and S were continuously cultivated under CK condition for two additional growth cycles under control conditions (P2 and P3, Figure 1B). '' (In method)
Comment 3: By RNA sequencing and WGCNA analysis, the key gene nodes related with the developmental growth were identified. However, the candidate genes seemed to be too numerous. The key genes related with TGP have not be focused. In the above trait variation analysis part, the content of Chl a, Chl b, F0, Fm, Fv/Fm, proline, soluble sugar, soluble protein, the activity of antioxidant enzymes, including APX, CAT, SOD, and POD have be demonstrated having close association with TGP for offspring resilience to MaE in duckweed. I suggest the authors to focus on the genes involved the synthesis or metabolic cycles of the above genes, to make the study having more coherence.
Response 3: We agree with your perspective. In many studies, concentrating on a select few genes can markedly augment the research's value, especially in articles that focus on significant traits. However, in this study, Transgenerational plasticity (TGP) is a complex trait that cannot be accomplished by the action of a single gene or a few genes alone. Therefore, compared to focusing solely on a few genes, we are more interested in the recruitment of Transcriptional memory genes (TMGs) during the transgenerational process, as this may better elucidate the occurrence of TGP and establish potential associations between molecular traits and phenotypic traits. Through weighted gene co-expression network analysis (WGCNA), we found that TMGs tend to be integrated into the same modules, indicating that they are indeed recruited together in the TGP process. Hub genes in network modules are considered to play a key role in the formation of module topology and have a higher weight of importance in the functions in which the module is involved. Therefore, by identifying hub genes in the network, we can also identify which TMGs are more important in the TGP process, thereby focusing the results more precisely.
Following your suggestion to enhance the coherence of the study, we have compiled information on Hub genes in key modules. According to our criteria, a total of 359 Hub genes were identified, of which 114 are TMGs, accounting for 32.59%. These TMGs are involved in processes such as development, response to oxidative stress, epigenetic processes, and photosynthesis.
For traits related to TGP, the transgenerational plasticity of phenotypic traits is likely the result of transcriptional memory of development-related genes. The transgenerational plasticity of antioxidant enzyme activity is likely associated with the transcriptional memory of genes related to antioxidant enzymes and oxidative stress. As for other traits, such as soluble sugar and protein, their phenotypic formation may be more complex. Therefore, in this article, we focused on genes related to development and oxidative stress resistance.
We have added Supplementary Table 5 to provide detailed information on hub genes, and added the following paragraph to the ''Results'' section to illustrate the number of TMGs among the hub genes and their functional characteristics:
''The top 10% of nodes by degree in each module were defined as hub genes. According to this criterion, a total of 359 hub genes were identified in both the Weaken-traits related and Strengthen-traits related modules. Categorized by the biological processes they participate in, 94 are associated with growth and development, 12 with epigenetic processes, 12 with the response to oxidative stress, 26 with photosynthesis, and 45 with the response to light stimulus. The Weaken-traits related modules (turquoise, purple, magenta, lightgreen, and green) contain 39 TMG2 and 52 TMG3, most of which are primarily associated with growth and development. The Strengthen-traits related modules (yellow, salmon, red, lightcyan, and blue) contain 18 TMG2 and 5 TMG3, most of which are primarily associated with functions related to the response to oxidative stress. Therefore, TMGs account for 32.59% of the hub genes. The information of hub genes was listed in Supplementary Table 5. ''
Comment 4: 8 genes are not enough for validation of gene expression revealed by RNA sequencing. The expression pattern of CK were all 1.0 level from Figure 9 (A to H). The expression pattern of CK can be removed. The expression patterns of all selected TMG3 genes from transcriptome and qRT-PCR can be put in one figure, to compare their expression trend.
Response 4: We fully concur with your viewpoint. However, the experimental materials for this study have been depleted, making it quite challenging to augment the number of qRT-PCR samples. In Figure 9, we aim not only to validate the reliability of the RNA sequencing results but also to demonstrate the transcriptional memory of gene expression underlying transgenerational plasticity, which is reflected by the maintenance of original expression patterns in some genes even as the environment shifts from S treatment to CK. Given the evident capacity for transgenerational plasticity exhibited by duckweed, we believe that adding more qRT-PCR samples would not alter our conclusions.
In this figure, CK being set to 1 is one of the commonly used analytical methods (e.g. https://doi.org/10.3390/ijms252011086). It is obtained through normalization with the aim of allowing readers to better compare the differences between treated and control groups. Following your suggestion, we have included the FPKM values obtained from the sequencing results in the figure, which facilitates readers to compare their expression trends.
Comment 5: The resolution of figure 5 (C to F) and figure 6 (C and D) are very low, which are fuzzy. The dpi of figures should be above 300.
Response 5: Sorry for the fuzzy figure. In fact, we have provided high-resolution images, but due to the compression of image quality by the review system, they may appear blurry. However, this issue should not arise in the final publication.
Round 2
Reviewer 1 Report
Comments and Suggestions for Authors
Transgenerational Plasticity Enhances the Tolerance of Duckweed (Lemna minor) to Stress from Exudates of Microcystis aeruginosa
Gengyun Li 1,2, Tiantian Zheng 1, Gang Wang 1, Qian Gu 1, Xuexiu Chang 3,4, Yu Qian 1, Xiao Xu 1, Yi Wang 1, Bo Li 1,* and Yupeng Geng 1,*
The authors replied to my critique that the species was not identified as Lemna minor with the information who identified it. This is not satisfying at all, especially as this species is not very common in China. Without molecular methods (I mentioned already two methods that are required) it is not clear which species we are talking about.
In the R1 version of the manuscript, the authors also gave the cultivation conditions. As nutrient medium they mentioned HGZ-145. I do not know this medium; no reference was given. I suggest to give the composition of the medium here under “Material and methods”.
I suggest Major Revision.

Author Response
Comments 1: The authors replied to my critique that the species was not identified as Lemna minor with the information who identified it. This is not satisfying at all, especially as this species is not very common in China. Without molecular methods (I mentioned already two methods that are required) it is not clear which species we are talking about.
In the R1 version of the manuscript, the authors also gave the cultivation conditions. As nutrient medium they mentioned HGZ-145. I do not know this medium; no reference was given. I suggest to give the composition of the medium here under “Material and methods”.
I suggest Major Revision.
Response:
Thank you very much for your valuable comments. Below are our detailed responses:
1. Regarding the identification of Lemna minor
We fully understand the importance of accurate species identification in scientific research and appreciate your suggestion to conduct molecular identification. In response to your valuable feedback, we have initiated barcoding analysis of the Lemna minor material used in this study. However, due to the review system's requirement to respond within one week, the molecular identification results will be uploaded to the system once the analysis is completed.
The current duckweed classification system was established before the maturity of molecular technology, with morphological classification features being the main basis for identifying L. minor. According to the morphological descriptions by Bog et al. (2020), we confirm that the plant material used in this study is highly consistent with the morphological characteristics of L. minor. In fact, according to the description in "Flora of China," L. minor is a widely distributed species in China, commonly found in lakes, ponds, ditches, and other still or slow-flowing water bodies, often in suboceanic cool-to-moderately temperate climatic zones. Our team has used L. minor collected from the wild in other studies and conducted barcoding identification to confirm the accuracy of the materials. However, the L. minor materials in this study were not directly collected from the wild but were purchased from the germplasm resource library of the Chengdu Institute of Biology, Chinese Academy of Sciences. This institution has established a national aquatic germplasm resource library, and all the germplasm resources included have been identified by authoritative taxonomists before being stored. Many researchers have published numerous related papers using these resources. Therefore, based on the recognition of the authority of the national germplasm resource library, we did not perform additional barcodes, but we agree with the reviewer's suggestion, and the barcode work is already underway.
Finally, it is important to emphasize that the primary focus of our study is on transgenerational plasticity, a mechanism widely observed in plants. Our experimental results demonstrate a significant effect of this phenomenon, which is not influenced by potential uncertainties in species identification.
2. Regarding the HGZ-145 nutrient medium
Thank you for pointing out that the composition of the HGZ-145 nutrient medium was not clearly described in the initial manuscript. In the revised version, we have included a reference (numbered 46) that provides a detailed description of the composition and formulation of the HGZ-145 nutrient medium.
Reviewer 2 Report
Comments and Suggestions for Authors
The author has revised the manuscript according to the comments. It could be published now.
Author Response
Comments1: The author has revised the manuscript according to the comments. It could be published now.
Response: Thank you very much.
Round 3
Reviewer 1 Report
Comments and Suggestions for Authors
I understand the difficulties of the authors very well but do they really want to publish a manuscript where the identity of the plant species is uncertain? I agree that the system of taxonomy of duckweed was developed long before methods of molecular taxonomy were developed and I also know the cited paper of Bog et al. Classical methods are still important because one cannot carry out molecular methods with each sample collected in nature and a preliminary identification is important. However, for a published manuscript the identity of the investigated species must be clear.
When you need more time for required experiments to improve the quality of the manuscript, the journal must give you this time.
I ask the authors to read the last three papers of Laura Morello, Milan about the method of tubulin-based polymorphism and they will agree that molecular methods are required.
H. Xu et al. from CAS in Chengdu, where the samples in the present manuscript came from, published a paper in Mol. Biol. Rep. (2012) 39: 547–554 and listed in Table 1 16 Lemna samples and identified between them six different Lemna species. On this basis they came to the unacceptable conclusion that geographic distribution is more important for the genetic diversity than the species identity. Late Elias Landolt, ETH Zurich investigated these species also (on the basis of morphology) and identified all 16 samples as Lemna aequinoctialis. However, Landolt was the best specialist of duckweed morphology and I do not know one living specialist who can distinguish all 35 duckweed species (and 2 hybrids). Therefore, I suggest urgently to use molecular methods for identification of Lemna minor, which is a representative of the most difficult duckweed group. The experimental effort is only small.
Moreover, did the authors give the light intensity in micromol m-2 s-1?
Author Response
Suggestions for Authors
I understand the difficulties of the authors very well but do they really want to publish a manuscript where the identity of the plant species is uncertain? I agree that the system of taxonomy of duckweed was developed long before methods of molecular taxonomy were developed and I also know the cited paper of Bog et al. Classical methods are still important because one cannot carry out molecular methods with each sample collected in nature and a preliminary identification is important. However, for a published manuscript the identity of the investigated species must be clear.
When you need more time for required experiments to improve the quality of the manuscript, the journal must give you this time.
I ask the authors to read the last three papers of Laura Morello, Milan about the method of tubulin-based polymorphism and they will agree that molecular methods are required.
H. Xu et al. from CAS in Chengdu, where the samples in the present manuscript came from, published a paper in Mol. Biol. Rep. (2012) 39: 547–554 and listed in Table 1 16 Lemna samples and identified between them six different Lemna species. On this basis they came to the unacceptable conclusion that geographic distribution is more important for the genetic diversity than the species identity. Late Elias Landolt, ETH Zurich investigated these species also (on the basis of morphology) and identified all 16 samples as Lemna aequinoctialis. However, Landolt was the best specialist of duckweed morphology and I do not know one living specialist who can distinguish all 35 duckweed species (and 2 hybrids). Therefore, I suggest urgently to use molecular methods for identification of Lemna minor, which is a representative of the most difficult duckweed group. The experimental effort is only small.
Moreover, did the authors give the light intensity in micromol m-2 s-1?
Response:
Thank you for the reviewer’s comments. We fully agree with the reviewer that the plant material must be accurately identified. In fact, after receiving the first-round review comments, we began the barcoding experiment, and the results were only recently obtained. After amplifying the sequences using the atpF-atpH and psbK-psbI primer pairs, we performed a blastn comparison of the obtained sequences. Considering metrics such as Query Cover, Percent Identity, and Total Score, the results showed that the strain most closely related to Lemna minor strain Skf_B & C, with both primer pairs yielding 100% percent identity, and the query cover values were 93% (atpF-atpH) and 94% (psbK-psbI), respectively. We noted that some studies mention that using barcoding to distinguish between the sister species L. minor and L. japonica is not always straightforward. Our blast results show that although L. japonica did obtain a relatively high score, it was lower than that for L. minor, and the closest sequences for L. japonica came from different strains, unlike L. minor, which corresponded to the same strain. We have uploaded the blast results as Supplementary Figure S1. Additionally, based on the taxonomic descriptions of L. minor and L. japonica by Bog et al. (2020), we note some significant phenotypic differences between the two species. Based on the barcoding results and the consistent phenotypic descriptions of Lemna minor by Bog et al. (2020), we confirm that the plant material used in this study is indeed Lemna minor.
The reviewer also suggested using the Tubulin-based Polymorphism (TBP) method for further validation. However, our laboratory currently lacks the necessary equipment, and research has shown that TBP exhibits substantial intraspecific polymorphism between different strains of Lemna minor (Braglia et al., 2021). In other words, even if our material’s TBP amplification results differ from those reported in the literature, it cannot definitively determine whether the material is Lemna minor. Therefore, we have decided not to perform further TBP experiment.
We have added descriptions of barcoding identification and light intensity in the 'Materials and Methods' section. The specific changes are as follows:
'Barcoding was employed to further verify the species identity of the plant material, following the methodologies described in previous studies [47,48]. In brief, DNA was extracted using the FastPure Plant DNA Isolation Mini Kit (Vazyme, Nanjing, China), and plastid intergenic spacers were amplified with the atpF-atpH primers (Forward: 5'-ACTCGCACACACTCCCTTTCC-3', Reverse: 5'-GCTTTTATGGAAGCTTTAACAAT-3') and psbK-psbI primers (Forward: 5'-TTAGCATTTGTTTGGCAAG-3', Reverse: 5'-AAAGTTTGAGAGTAAGCAT-3'). Sanger sequencing was performed, and the sequences were subjected to blastn alignment. The results confirmed that the obtained sequences were most closely related to L. minor (Supplementary Figure S1). The strain was clonally propagated for genetic consistency and cultured in HGZ-145 nutrient solution [46] in a climate-controlled chamber at 24-26°C, with light intensity of 2200-2500 lux (approximately 41-46 µmol m⁻² s⁻¹) and a 16:8 light-dark cycle.'
Accordingly, two additional references have been included:
- Braglia, L.; Lauria, M.; Appenroth, K.J.; Bog, M.; Breviario, D.; Grasso, A.; Gavazzi, F.; Morello, L. Duckweed Species Genotyping and Interspecific Hybrid Discovery by Tubulin-Based Polymorphism Fingerprinting. Front. Plant Sci. 2021, 12, 625670, doi:https://doi.org/10.3389/fpls.2021.625670.
- Braglia, L.; Ceschin, S.; Iannelli, M.A.; Bog, M.; Fabriani, M.; Frugis, G.; Gavazzi, F.; Gianì, S.; Mariani, F.; Muzzi, M.; et al. Characterization of the Cryptic Interspecific Hybrid Lemna×mediterranea by an Integrated Approach Provides New Insights into Duckweed Diversity. J. Exp. Bot. 2024, 75, 3092–3110, doi:https://doi.org/10.1093/jxb/erae059.